# Tetraspanin 5 (TSPAN5), a Novel Gatekeeper of the Tumor Suppressor DLC1 and Myocardin-Related Transcription Factors (MRTFs), Controls HCC Growth and Senescence

**DOI:** 10.3390/cancers13215373

**Published:** 2021-10-26

**Authors:** Laura Schreyer, Constanze Mittermeier, Miriam J. Franz, Melanie A. Meier, Dietmar E. Martin, Kerstin C. Maier, Kerstin Huebner, Regine Schneider-Stock, Stephan Singer, Kerstin Holzer, Dagmar Fischer, Silvia Ribback, Bernhard Liebl, Thomas Gudermann, Achim Aigner, Susanne Muehlich

**Affiliations:** 1Department of Chemistry and Pharmacy, Friedrich-Alexander-Universität Erlangen-Nürnberg, 91058 Erlangen, Germany; laura.ls.schreyer@fau.de (L.S.); miriam.franz@fau.de (M.J.F.); melanie.a.meier@fau.de (M.A.M.); dagmar.fischer@fau.de (D.F.); 2Cancer Science Institute of Singapore, National University of Singapore, Singapore 117599, Singapore; constanze.mittermeier@nus.edu.sg; 3Gene Center, Department of Chemistry and Pharmacy, Ludwig-Maximilians-University Munich, 81377 Munich, Germany; martin@genzentrum.lmu.de (D.E.M.); kerstin.maier@mpibpc.mpg.de (K.C.M.); 4Experimental Tumor Pathology, Institute of Pathology, University Hospital, Friedrich-Alexander-Universität Erlangen-Nürnberg, 91054 Erlangen, Germany; kerstin.huebner@uk-erlangen.de (K.H.); regine.schneider-stock@uk-erlangen.de (R.S.-S.); 5Department for Pathology, University Hospital Tuebingen, 72076 Tuebingen, Germany; stephan.singer@med.uni-tuebingen.de (S.S.); kerstin.singer@med.uni-tuebingen.de (K.H.); 6Institute for Pathology, University of Greifswald, 17475 Greifswald, Germany; silvia.ribback@uni-greifswald.de; 7LGL Bayerisches Landesamt für Gesundheit und Lebensmittelsicherheit, 85764 Oberschleißheim, Germany; Liebl@lgl-bayern.de; 8Walther Straub Institute of Pharmacology and Toxicology, Ludwig-Maximilians-University Munich, 80336 Munich, Germany; thomas.gudermann@lrz.uni-muenchen.de; 9Rudolf Boehm Institute of Pharmacology and Toxicology, Clinical Pharmacology, University of Leipzig, 04107 Leipzig, Germany; achim.aigner@medizin.uni-leipzig.de

**Keywords:** DLC1, TSPAN5, MKL1, MRTF, SRF

## Abstract

**Simple Summary:**

Hepatocellular carcinoma (HCC) ranks second among the leading causes of cancer-related death. Since current therapeutic options are very limited, a deeper understanding of the molecular mechanisms underlying the tumor onset and progression of HCC holds great potential for improved therapeutic options. Although it has been shown that deleted in liver cancer 1 (DLC1) acts as a tumor suppressor whose allele is lost in 50% of liver cancers, alterations in gene expression initiated by DLC1 loss have not yet been the primary focus of liver cancer research. To identify novel gene targets that allow for a personalized medicine approach for HCC therapy, we performed gene expression profiling for HepG2 cells stably expressing DLC1shRNA. We provide evidence that TSPAN5 is required for HCC growth, migration and invasion, and dissected the underlying molecular mechanisms involving myocardin-related transcription factors. Thus, TSPAN5 represents a novel therapeutic target for the treatment of HCC characterized by DLC1 loss.

**Abstract:**

Human hepatocellular carcinoma (HCC) is among the most lethal and common cancers in the human population, and new molecular targets for therapeutic intervention are urgently needed. Deleted in liver cancer 1 (DLC1) was originally identified as a tumor suppressor gene in human HCC. DLC1 is a Rho-GTPase-activating protein (RhoGAP) which accelerates the return of RhoGTPases to an inactive state. We recently described that the restoration of DLC1 expression induces cellular senescence. However, this principle is not amenable to direct therapeutic targeting. We therefore performed gene expression profiling for HepG2 cells depleted of DLC1 to identify druggable gene targets mediating the effects of DLC1 on senescence induction. This approach revealed that versican (VCAN), tetraspanin 5 (TSPAN5) and N-cadherin (CDH2) were strongly upregulated upon DLC1 depletion in HCC cells, but only TSPAN5 affected the proliferation of HCC cells and human HCC. The depletion of TSPAN5 induced oncogene-induced senescence (OIS), mediated by the p16^INK4a^/pRb pathways. Mechanistically, silencing TSPAN5 reduced actin polymerization and thereby myocardin-related transcription factor A- filamin A (MRTF-A-FLNA) complex formation, resulting in decreased expression of MRTF/SRF-dependent target genes and senescence induction in vitro and in vivo. Our results identify TSPAN5 as a novel druggable target for HCC.

## 1. Introduction

Hepatocellular carcinoma (HCC) represents the sixth most common cancer and the second leading cause of cancer deaths worldwide [1]. The etiology of HCC involves hepatitis B (HBV) and hepatitis C (HCV) viral infection, type-2-diabetes-associated steatohepatitis, alcohol consumption and environmental carcinogens such as aflatoxin B1 contamination [2,3,4]. These various insults can result in chronic liver injury and liver cirrhosis, considered to be the precursor of HCC [5]. In the preneoplastic cirrhosis stage, the induction of cellular senescence protects from malignant transformation [6]. Owing to current paucity of drugs for HCC therapy, there is an urgent need to unravel new therapeutic intervention strategies in a personalized medicine approach targeting the molecular mechanisms of HCC development. 

On the genetic level, deleted in liver cancer 1 (DLC1) is a tumor suppressor gene, whose allele was found to be lost in about 50% of liver cancers [7]. Previously, we described that DLC1 expression induces senescence in HCC cells [8]. The induction of senescence acts as a tumor-suppressive mechanism, and may therefore hold promise for pharmacological intervention in HCC therapy. 

Although the importance of DLC1 loss for liver tumorigenesis is well established, the molecular mechanisms by which DLC1 loss exerts its effects on the senescence response remain to be investigated. 

Here, we carried out microarray analyses in HepG2 DLC1 knockdown cells and found expression of tetraspanin 5 (TSPAN5), versican (VCAN) and N-cadherin (CDH2) strongly induced. Among these, we identified only TSPAN5 to have a pivotal effect on proliferation of HCC cells and thus HCC growth. Tetraspanins form a widely distributed protein superfamily with four transmembrane domains, as well as two extracellular and one intracellular loops, but do not function as direct receptor molecules [9]. Tetraspanins organize themselves with other membrane molecules into so-called tetraspanin-enriched microdomains (TEMs) [10]. Within these TEMs, interactions between tetraspanins and the actin cytoskeleton occur, affecting cell adhesion, migration, invasion and signaling [11,12,13]. Here, we report that TSPAN5 interferes with actin polymerization and complex formation of myocardin-related transcription factor A (MRTF-A) and the actin-binding protein filamin A (FLNA), which is essential for MRTF transcriptional activity. As a result, depletion of TSPAN5 reduces MRTF/SRF-dependent target genes and induces p16^INK4a^/pRb pathways, leading to oncogene-induced senescence (OIS) in vitro and in vivo. Our results open new therapeutic approaches in DLC1-deficient HCC by using TSPAN5 as a new drug target.

## 2. Materials and Methods

### 2.1. Cell Culture, Transfections and Reagents

HuH6, Hep3B and 3T3 cells were cultured in Dulbecco’s Modified Eagle’s Medium (DMEM; Sigma-Aldrich, Taufkirchen, Germany), A7 and M2 cells in Eagle’s Minimum Essential Medium (MEM; Sigma-Aldrich, Taufkirchen, Germany) and HuH7, HepG2 and HepG2 CRISPR/Cas9 DLC1 KO (HepG2 DLC1 KO) or HepG2 DLC1 wt cells in RPMI 1640 medium (Sigma-Aldrich, Taufkirchen, Germany), all supplemented with 10% fetal bovine serum (FBS; Invitrogen, Karlsruhe, Germany) and 1% penicillin/streptomycin (Sigma-Aldrich, Taufkirchen, Germany). The cell lines are listed in Appendix A. For transient transfections of siRNA and plasmids, lipofectamine RNAiMAX and lipofectamine 2000 (Invitrogen, Karlsruhe, Germany) were used according to the manufacturer’s instructions. The sequences of the siRNAs and the plasmids used in this study are listed in Appendix A.

### 2.2. Cell Proliferation Assay

HCC cells were seeded and transfected with the appropriate siRNA or plasmids, as described above. Twenty-four hours after transfection, the cell count for day 1 is determined using a Neubauer counting chamber. On the following days, the cell count of a further well is determined analogously at 24 h intervals. For each cell count, the corresponding well is first carefully washed with PBS and trypsinized, and 10 µL of the homogeneous cell suspension is then transferred to the Neubauer counting chamber to calculate the cell count. 

### 2.3. Senescence-Associated ß-Galactosidase Staining

The cellular senescence of treated cells was assayed after 5 days by using a senescence ß-galactosidase staining kit according to the manufacturer’s instructions (Cell Signaling Technology, Danvers, MA, USA). The percentage of SA-β-gal-positive cells was calculated by counting the blue-stained cells. 

### 2.4. Scratch Wound Assay

Cells were seeded at a density sufficient to obtain a confluent monolayer. After scratching a wound the migrated cells were counted, and the width of the wound was measured regularly.

### 2.5. Invasion Assay

The cell invasion assay was performed using Corning BioCoat Matrigel Invasion Chambers (Corning, Tewksbury, MA, USA) according to the manufacturer’s instructions. After incubation for 22 h at 37 °C, invaded cells were methanol-fixed, stained with crystal violet and counted under a light microscope (Carl Zeiss, Oberkochen, Germany).

### 2.6. Spheroid Assay

Spheroid assays were performed by using Kugelmeiers Sphericalplate 5D^®^ (Kugelmeiers, Erlenbach, Germany) according to the manufacturer’s instructions. The Sphericalplate^®^ (Kugelmeiers, Erlenbach, Germany) was incubated at 37 °C in a CO_2_ atmosphere for 11 days. The spheroid formation and growth were recorded daily and captured in images via optical microscopy (Carl Zeiss, Oberkochen, Germany).

### 2.7. RNA Extraction, cDNA Synthesis and Quantitative Real-Time PCR Analysis (qRT-PCR)

The total RNA of the in vitro experiments was isolated using TRIzol Reagent (Merck, Darmstadt, Germany). The RNA of the in vivo models was isolated and purified using an AllPrep DNA/RNA/Protein Mini Kit (Qiagen, Hilden, Germany) according to the manufacturer’s instructions. One microgram of RNA was reverse-transcribed into cDNA with SuperScript II Reverse Transcriptase (Merck, Darmstadt, Germany). qRT-PCR was carried out with SYBR Green I (Roche, Mannheim, Germany) and gene-specific primers (listed in Appendix A) in a LightCycler 96 (Roche, Mannheim, Germany). To normalize the mRNA levels for equal loading, 18S rRNA was used. 

### 2.8. Primer Synthesis

Gene-specific primers for qRT-PCR and ChIP were designed using a Roche Universal Probe Library/Assay Design Center and purchased from Metabion (Martinsried, Germany). The primer sequences are listed in Appendix A.

### 2.9. Microarray Analysis

Microarray analysis was performed and examined as reported previously [14]. The used kits are listed in Appendix A.

### 2.10. Chromatin Immunoprecipitation (ChIP)

Cells were stimulated with 20% FBS (Merck, Darmstadt, Germany) for 20 min. Cross-linking was executed with 1% formaldehyde and quenching with 0.125 M glycine. One hundred micrograms of sheared chromatin was immunoprecipitated with antibodies (listed in Appendix A) coupled to Dynabeads Protein G (Life Technologies, Darmstadt, Germany) overnight at 4 °C. For input, 1% of sonicated chromatin without antibodies was set aside. qRT-PCR was performed with the eluted DNA (RNeasy Mini Kit; Qiagen, Hilden, Germany) using specific primers for promoters of TSPAN5, VCAN and CDH2 (listed in Appendix A).

### 2.11. RhoA Assay

Immunoprecipitation was performed with anti-active RhoA antibodies (listed in Appendix A). Before immunoprecipitation, cells were rinsed in ice-cold phosphate-buffered saline (PBS) and lysed in 500 µL immunoprecipitation buffer. After 45 min incubation on ice, lysates were centrifuged at 12,000× *g* for 15 min at 4 °C. After overnight incubation with anti-active RhoA antibodies, 100 µL of recombinant Dynabeads Protein G (Life Technologies, Darmstadt, Germany) in an immunoprecipitation buffer was added and the lysates were rotated for 4 h at 4 °C. Immunoprecipitates were washed four times with an immunoprecipitation buffer and then resolved by sodium dodecyl sulfate-polyacrylamide gel electrophoresis (SDS-PAGE). Proteins were then transferred to PVDF membranes and immunoblotted with the anti-RhoA polyclonal antibody.

### 2.12. Immunoblotting

Proteins were separated according to molecular weight by electrophoresis with an appropriate percent SDS gel for approximately 2 h at 100 V (SDS-PAGE). Subsequently, the proteins were transferred at 350 mA for approximately 1.5 h onto a polyvinylidene fluoride (PVDF) membrane (Merck, Darmstadt, Germany) previously activated in 100% methanol and equilibrated in a transfer buffer. To avoid non-specific protein binding, the PVDF membrane is thereafter blocked in 5% milk powder in Tris-buffered saline with Tween20 (TBS-T) for 1 h at room temperature and gently shaken. After washing the membrane thrice with TBS-T, the membrane is incubated in a dilution of the appropriate primary antibody (listed in Appendix A) overnight at 4 °C with gentle agitation. After being washed another three times with TBS-T, on the next day the membrane was incubated with horseradish-peroxidase-coupled secondary antibodies for 1 h at room temperature and gently agitated. Finally, the proteins were detected by chemiluminescence in a luminescent imager (ChemiDoc Imaging System, Bio-Rad Laboratories, Hercules, CA, USA).

### 2.13. Immunofluorescence Staining

Cells were fixed with 4% paraformaldehyde/PBS for 10 min and permeabilized with 0.2% Triton X-100/PBS for 7 min, both at room temperature, prior to blocking unspecific binding with 1% BSA/PBS for 30 min at 37 °C. After incubation with the primary antibody (PML antibody; Santa Cruz Biotechnology, Santa Cruz, CA, USA) for 1 h at room temperature, the cells were incubated with the secondary antibody labeled with Alexa Fluor 488 (Merck, Darmstadt, Germany) for 30 min. F-actin filaments were stained with phallotoxin A555 (Cytoskeleton, Denver, CO, USA) and focal adhesions with paxillin (BD Bioscience, San Jose, CA, USA). Nuclear staining was performed using DAPI (Sigma-Aldrich, Darmstadt, Germany). Images were taken by confocal microscopy (Carl Zeiss, Oberkochen, Germany; Leica Camera, Wetzlar, Germany).

### 2.14. Measuring the Focal Adhesion Length

Focal adhesions were stained with anti-paxillin antibodies as described above. Pictures of the layer containing the focal adhesions were taken by confocal microscopy and the length of the focal adhesion was measured manually using the ImageJ software (Wayne Rasband, National Institutes of Health, Bethesda, MD, USA). The focal adhesion lengths of at least five randomly selected fields were used for analysis. 

### 2.15. In Situ Proximity Ligation Assay (PLA)

An in situ proximity ligation assay was performed and detected as described previously [15]. The primary antibodies are listed in Appendix A.

### 2.16. Chicken Chorioallantoic Membrane Assay (CAM Assay) 

A chicken chorioallantoic membrane assay (CAM assay) was performed as described by Ribatti and colleagues [16]. Fertilized, specific-pathogen-free (SPF) chicken eggs (VALO Biomedia, Osterholz-Scharmbeck, Germany) were incubated at 37 °C and the egg was opened on day 8 of embryonic development. For this, a window of approximately 2 cm was cut into the shell of the broad pole of the chicken eggs and sealed with tape (Durapore silk tape, 3M, Saint Paul, MN, USA). The next day, 1.0 × 10^6^ HCC cells were embedded and pelleted in growth-factor-reduced Matrigel (Corning, Wiesbaden, Germany) and then transplanted onto the CAM. Normal HuH7 cells and a more aggressive HepG2 cell clone (“clone 5”; Institute of Experimental Tumor Pathology, Friedrich-Alexander University Erlangen [17]) were used. After an additional 5 days of incubation, the tumor with the surrounding CAM was fixed with 4% formaldehyde, excised and measured. Tumors were then either embedded in paraffin and cut into 3–5 µm sections for immunhistochemical evaluation or cryopreserved in liquid nitrogen for gene and protein expression analysis.

### 2.17. Immunohistochemistry (IHC)

Formalin-fixed and paraffin-embedded sections (1–3 µm) of excised CAM tumors were deparaffinized with xylene prior to rehydradation with graded ethanol according to standard procedures. Validated IHC protocols established at the Institute of Pathology of FAU Erlangen were applied for hematoxylin and eosin (HE), Ki67 and p16 staining. For this purpose, an IHC apparatus “Ventana BenchMark Ultra instrument” and an “UltraView Universal DAB Detection Kit” (both Ventana Medical Systems, Oro Valley, AR, USA) were used according to the manufacturer’s instructions. TSPAN5, MRTF-A and H3K9me3 were stained manually. After antigen-specific, heat-induced antigen retrieval, endogenous peroxidases were first blocked, slides were incubated with the primary antibody (listed in Appendix A) overnight at RT and then incubated with the biotinylated secondary antibody. Sections were detected with a VECTASTAIN^®^ Elite^®^ ABC Kit (Vector Laboratories, Burlingame, CA, USA) and DAB substrate (Dako, Santa Clara, CA, USA), and counterstained with hematoxylin.

### 2.18. Scanning of IHC Sections

Prepared and stained IHC sections of CAM tumors were scanned using a panoramic MIDI system (camera type: CIS VCCFC60FR19CL; objective: plan-apochromat; magnification: 40×; and camera adapter: PanoramicMidi and Panoramic Flash 250, magnification: ×1, 3DHISTECH, Budapest, Hungary). The scans were viewed and analyzed using CaseViewer software (3DHISTECH, Budapest, Hungary).

### 2.19. Human Liver Samples

Non-tumorous and tumorous liver samples were obtained from routine diagnostic surgical specimens sent to the Pathology Institute of the Greifswald Medical School. Ethical approval was obtained from the local Ethics Committee of the Greifswald Medical School (no. BB67/10). Patient characteristics including age, gender, grading, staging, lymphangiosis and hemangiosis carcinomatosa, R-status and tumor size are listed in Appendix A.

### 2.20. Subcutaneous Tumor Xenograft

The treatment of subcutaneous tumor xenograft mice to study the in vivo effects of TSPAN5 knockdown was essentially performed according to protocols described previously [8,18,19,20,21,22]. Animal studies complied with the ARRIVE guidelines, were carried out in accordance with the EU Directive 2010/63/EU for animal experiments with national regulations and were approved by the local authorities (Landesdirektion Sachsen). Mice were kept in cages with rodent chow (ssniff, Soest, Germany) and water available ad libitum. For the generation of s.c. tumor xenografts, 2 × 10^6^ Huh7 cells in 150 µL PBS were subcutaneously injected into athymic nude mice (Crl:CD1-Foxn1nu, Charles River Laboratories, Sulzfeld, Germany). Two and a half weeks after injection, mice were randomized into negative control and specific treatment groups. PEI F25-LMW/siRNA complexes containing 10 μg of siRNA in 150 μL of HN buffer were intraperitoneally (i.p.) injected 3× per week. On day 10 after treatment start, mice were sacrificed, the tumors were excised and shock-frozen in liquid nitrogen for subsequent RNA and protein preparation.

### 2.21. Statistical Analysis

Statistical analysis was performed using a two-tailed unpaired Student’s *t*-test or one-way ANOVA, followed by a Tukey´s multiple comparison test. Unless stated otherwise, experimental data are derived from three independent experiments. Values are given as the mean ± standard deviation (mean ± SD). Values are considered significant, with *, *p* < 0.05; **, *p* < 0.01; and ***, *p* < 0.001.

## 3. Results

### 3.1. Gene Expression Profiling to Identify Novel Target Genes Induced by DLC1 Loss 

To identify novel target genes mediating the effect of DLC1 loss on HCC growth and the senescence response, we performed microarray experiments in HepG2 human hepatoma cells using Affymetrix oligonucleotide arrays for differential expression studies. For stable DLC1 knockdown, cells were transduced with DLC1 shRNA for RNAi induction. We selected the top five genes from the list of 48 genes that were upregulated in HepG2 DLC1 knockdown cells by a factor of at least 3.2 compared to the control cells, and identified five novel DLC1-dependent genes: versican (VCAN), tetraspanin 5 (TSPAN5), meprin 1A (MEP1A), histon cluster 1 H2B family member K (HIST1H2BK) and N-cadherin (CDH2) (Figure 1A). A complete list of the 48 differentially expressed target genes is shown in Appendix A. We were able to confirm all five genes, via real-time PCR, to be upregulated in HepG2 expressing DLC1 shRNA (Appendix A), as well as in HepG2 and Hep3B cells after transient DLC1 knockdown via siRNA transfection (Figure 1B and Appendix A). This upregulation of VCAN, TSPAN5 and CDH2 was also observed on the protein level (Figure 1C and Appendix A). Vice versa, all genes except for HIST1H2BK were downregulated upon DLC1 re-expression in HuH7 HCC cells (Figure 1D). 

### 3.2. Transcriptional Regulation of the Newly Identified DLC1-Dependent Genes

We recently reported that the transcriptional coactivators MRTF-A/-B mediate the effects of DLC1 loss, and that the actin-binding protein filamin A (FLNA) is required for MRTF-A transcriptional activity [8,23,24]. We therefore investigated next whether silencing MRTF-A/-B or FLNA expression prevents the upregulation of the genes identified above. Indeed, co-transfection of siDLC1 with MRTF-A/-B siRNA or FLNA siRNA abolished the DLC1 knockdown-mediated upregulation of VCAN, TSPAN5 and CDH2 (Figure 2A). Furthermore, endogenous VCAN, TSPAN5 and CDH2 mRNA expression was downregulated upon MRTF-A/-B knockdown, while levels were rescued upon co-transfection of constitutively active MRTF-A (MRTF-A N100) (Figure 2B and Appendix A). Taken together, this demonstrates the role of MRTFs in VCAN and TSPAN5 expression. In order to analyze the direct recruitment of MRTF-A to the VCAN, TSPAN5 and CDH2 promoters, we performed chromatin immunoprecipitation (ChIP) with MRTF-A antibodies. We were able to amplify the VCAN, TSPAN5 and CDH2 promoters from MRTF-A immunoprecipitates in FLNA-positive A7 cells, but not from IgG immunoprecipitates as a negative control (Figure 2C). Moreover, the VCAN, TSPAN5 and CDH2 promoters were enriched only in FLNA-positive (A7) cells, but not in FLNA-negative (M2) cells. This indicates that VCAN, TSPAN5 and CDH2 are under the transcriptional control of MRTFs and FLNA, and may mediate the oncogenic effects of the MRTFs on HCC proliferation and migration upon DLC1 loss.

### 3.3. Requirement of TSPAN5 for HCC Proliferation, Migration, Invasion and Spheroid Formation

Since loss of the tumor suppressor DLC1 leads to increased HCC growth [8], we next investigated whether the newly identified target genes VCAN, TSPAN5 and CDH2 affect tumor characteristics such as proliferation or invasion. To unravel the target gene(s) responsible for HCC cell proliferation, we depleted VCAN, TSPAN5 and CDH2 via RNA interference and monitored cell numbers over time. While VCAN and CDH2 did not affect HuH7 or HuH6 cell proliferation, knockdown of TSPAN5 provoked a growth arrest (Figure 3A and Appendix A). In order to demonstrate that TSPAN5 is responsible for mediating the effects of DLC1 loss on HCC cell proliferation, a CRISPR/Cas9-based DLC1 KO HepG2 cell line was established. We observed enhanced cell proliferation in HepG2 DLC1 KO cells. Knockdown of TSPAN5 in HepG2 DLC1 KO cells reverted the proliferative effect of DLC1 loss, suggesting that TSPAN5 is able to mediate the effects of DLC1 loss on HCC cell proliferation (Figure 3B). Moreover, ectopic expression of the constitutively active MRTF-A variant (MRTF-A N100) partially prevented the TSPAN5-knockdown-induced proliferation arrest, indicating that pro-proliferative TSPAN5 effects function through MRTF-A (Figure 3C and Appendix A). Depletion of TSPAN5 significantly inhibited not only cell proliferation but also cell migration of HuH7 and HuH6 cells, as determined by a scratch wound healing assay (Appendix A). Knockdown of the other candidate target genes showed little (CDH2) or no effect (VCAN) on cell motility (Appendix A). Using a Transwell invasion assay based on the penetration of cells into 3D Matrigel along a gradient of conditioned medium, we found that the invasion of TSPAN5 knockdown cells was strongly reduced (Figure 3D). 

TSPAN5 depletion, but not VCAN or CDH2 depletion, rendered HuH7 cells unable to form spheroids (Figure 3E). These findings demonstrate the critical role of TSPAN5 in mediating critical hallmarks of hepatocarcinogenesis such as proliferation, invasion, migration and spheroid formation.

### 3.4. Requirement of TSPAN5 for Oncogene-Induced Senescence

We next sought to elucidate the molecular mechanisms underlying the TSPAN5-knockdown-mediated proliferation arrest. Given the transcriptional regulation of TSPAN5 by MRTFs as important regulators of the senescence response, we first examined a possible senescence induction upon TSPAN5 silencing. Senescence-associated ß-galactosidase (SA-ß-Gal) staining revealed the percentage of SA-ß-Gal-positive HuH7 and HuH6 cells to be strongly increased upon TSPAN5 depletion (Figure 4A and Appendix A). Since senescent cells manifest alterations in the cytoskeleton and the size of focal adhesions, we examined the actin cytoskeleton via F-actin and paxillin staining [25,26]. Characteristic morphological senescence-associated alterations, such as a reorganization of actin stress fibers and reduced focal adhesion length, revealed senescence induction as an underlying reason for the halting of proliferation in TSPAN5 knockdown cells (Figure 4B and Appendix A). To further prove that oncogene-induced senescence (OIS) causes the TSPAN5-knockdown-mediated growth arrest, we additionally assessed OIS markers. Firstly, we tested whether the oncogene Ras and the subsequent Ras-ERK1/2-p16^INK4a^ signaling cascade is activated after TSPAN5 loss by determining the phosphorylation status of ERK1/2 [27]. Indeed, enhanced phosphorylation of ERK1/2 was detected in TSPAN5-depleted HuH7 and HuH6 cells (Figure 4C and Appendix A). Secondly, we examined activation of the p16^INK4a^ tumor suppressor pathway, preventing phosphorylation of Rb via the CDK inhibitor p16^INK4a^, which in turn represses transcription factors necessary for cell cycle progression [28,29,30,31,32]. We detected an accumulation of p16^INK4a^ (Figure 4D) and a hypophosphorylation of Rb (Figure 4E and Appendix A). Thirdly, evaluation of senescence-associated heterochromatin foci (SAHF) demonstrated that TSPAN5-depleted HuH7 and HuH6 cells show an accumulation of histone H3 methylated on lysine 9 (Figure 4F and Appendix A) [33,34]. Fourthly, factors of the senescence-messaging secretome *CXCL10* and *TNFSF10* were significantly elevated upon TSPAN5 depletion in HuH7 and HuH6 cells (Figure 4G and Appendix A) [35,36,37]. Finally, we detected an accumulation of nucleolar promyelocytic leukemia (PML) bodies in TSPAN5-deficient cells via immunofluorescence (Figure 4H). Increased expression of PML leads to cell cycle arrest, and is therefore important for the senescent response [38,39,40]. Taken together, these results indicate that the knockdown of TSPAN5 exerts antiproliferative effects by inducing OIS. 

### 3.5. Depletion of TSPAN5 Affects the Actin/MRTF Signaling Axis

To reveal the mechanistic function of TSPAN5 on MRTF-A transcriptional activity, we first examined a putative link between TSPAN5, actin polymerization and the MRTF-A-FLNA complex. Since RhoA plays an important role in actin polymerization and MRTF-A transcriptional activation, we first assessed the RhoA activation status in HepG2 and HepG2 DLC1 KO cells [41]. As expected, RhoA activity was strongly increased in HepG2 DLC1 KO cells. To test whether TSPAN5 knockdown could cause a decrease in RhoA activity in HepG2 DLC1 KO cells, we transfected HepG2 DLC1 KO cells with TSPAN5 siRNA or control siRNA and performed RhoA activity assays. There was a decrease in RhoA activity in HepG2 DLC1 KO siTSPAN5 cells (Figure 5A), which could—as a master regulator of the actin cytoskeleton—affect actin polymerization. By actin fractionation, we indeed determined a decline in cellular F-actin content in HuH7 or HuH6 cells after TSPAN5 knockdown compared to control cells (Figure 5B and Appendix A). Considering that F-actin is required for the formation of the MRTF-A-FLNA complex [24], which is essential for MRTF/SRF target gene expression, we next examined MRTF-A-FLNA complex formation upon TSPAN5 depletion. Using proximity ligation assays, we found that the interaction between endogenous MRTF-A and FLNA was almost completely abolished in HuH7 or 3T3 cells upon TSPAN5 depletion (Figure 5C and Appendix A). Accordingly, mRNA and protein expression of MRTF-dependent target genes such as *SM22*, *GLIPR1* and *TGFß1* (Figure 5D and Appendix A) was significantly decreased. Depletion of TSPAN5 also impaired DLC1-knockdown-mediated *GLIPR1* induction, suggesting that TSPAN5 is responsible for mediating the effects of DLC1 loss on MRTF-dependent target gene expression (Appendix A). To ultimately test for the regulation of MRTF-dependent target genes by TSPAN5, we first depleted TSPAN5 by RNA interference and then reconstituted TSPAN5 expression by a siRNA-resistant form of TSPAN5. This GFP-TSPAN5 was able to rescue the inhibitory effect of TSPAN5 depletion on *SM22*, *GLIPR1* and *TGFß1* mRNA expression (Figure 5E). Collectively, these data indicate that actin polymerization, MRTF-A-FLNA complex formation and MRTF target gene expression are directly controlled by TSPAN5.

### 3.6. Anti-Tumor Effects of TSPAN5 Depletion in the Chorioallantoic Membrane (CAM) Assay

In order to provide evidence for the concept that TSPAN5 contributes to HCC growth via MRTF target genes, we first used a chorioallantoic membrane (CAM) assay. For this, TSPAN5 was knocked down in HuH7 cells or in aggressive HepG2 clone 5 cells [17], and cells were then transplanted onto a chicken CAM. Tumors excised after 5 days showed a significantly smaller volume after TSPAN5 depletion (Figure 6A and Appendix A). Efficient TSPAN5 knockdown was verified by gene expression analysis in the CAM model (Figure 6B). Importantly, reduced mRNA levels of the proliferation marker Ki67 indicated a significant decrease in mitosis and the proliferation rate (Figure 6B and Appendix A). This was also reflected by HE staining of CAM tumor sections. In HuH7 control cells, a very dense and aggressive growth was observed, with Matrigel being completely consumed and the CAM almost completely displaced by tumor cells. In some of the tumors, the extremely rapid growth of the tumor cells even led to necrosis, with typically recognizable pyknoses in the cell nuclei (Appendix A). In contrast, upon TSPAN5 knockdown the cells showed a looser and more scattered growth pattern with nests, indicating fewer tight cell–cell interactions with even partial disintegration of the tumor. Moreover, in line with the above in vitro results on tumor cell invasion, large parts of the Matrigel were still present and the CAM was intact. Immunohistochemical staining for Ki67 also revealed anti-proliferative effects of TSPAN5 knockdown, with loss of Ki67 immunopositivity in the HuH7 siTSPAN5 sections (Figure 6C left panel). Interestingly, immunohistochemical staining also revealed E-cadherin upregulation after TSPAN5 knockdown (Appendix A) [42,43,44]. Next, we investigated possible effects on MRTF target genes and the senescence response in the CAM model. Indeed, mRNA and protein expression of MRTF-dependent target genes such as SM22, GLIPR1, TGFß1, CTGF and CNN1 was strongly decreased upon TSPAN5 depletion (Figure 6D–F and Appendix A). In order to test whether the regression of CAM tumors upon TSPAN5 depletion is associated with OIS, we determined OIS markers such as ERK1/2 phosphorylation in HuH7 tumor lysates (Figure 6G). Additionally, we detected an accumulation of p16^INK4a^ and of histone H3 methylated on lysine 9 (H3K9me3) in immunohistochemical sections of HepG2 clone 5 and in HuH7 tumor lysates (Figure 6C right panel and Appendix A). These data indicated that TSPAN5 may be a valuable drug target with which to inhibit HCC growth by inducing OIS.

### 3.7. Anti-Tumor Effects of TSPAN5 Depletion in an HCC Xenograft Model

Collectively, our results suggest that TSPAN5 is a potential target in the treatment of DLC1-deficient HCC. To further substantiate the clinical relevance, we analyzed TSPAN5 expression in human HCC patient samples. All 11 patients with HCC showed elevated TSPAN5 protein levels compared to healthy individuals (Figure 7A). The patient characteristics are provided in Appendix A. The therapeutic efficacy of a TSPAN5 knockdown strategy in vivo was then evaluated in a tumor xenograft mouse model. Subcutaneous tumor xenografts were generated by injecting HuH7 cells into both flanks of athymic nude mice. For in vivo siRNA delivery, we used polymeric nanoparticles based on the complexation of the siRNA with the cationic low-molecular-weight polyethyleneimine PEI F25-LMW [45,46]. Upon the formation of xenograft tumors with solid growth kinetics, the mice were randomized into treatment (siTSPAN5) and negative control groups (sictrl) and treated systemically by an intraperitoneal (i.p.) injection of PEI/siRNA complexes three times per week. Importantly, xenograft growth and tumor volume were strongly reduced in the TSPAN5-specific treatment group (Figure 7B and Appendix A). 

The analysis of the excised xenograft tumors upon termination of the experiment confirmed efficient therapeutic TSPAN5 knockdown (Figure 7C). Reduced Ki67 mRNA indicated the inhibition of proliferation and mitotic rates in the siTSPAN5 treatment group as at least one underlying reason for the observed tumor inhibitory effects. Furthermore, MRTF-A target genes were found decreased in both mRNA and protein levels (Figure 7C–E). We next sought to examine whether therapeutic knockdown of TSPAN5 is capable of inducing oncogene-induced senescence in vivo. 

The analysis of the phosphorylation status of ERK1/2 and retinoblastoma protein (Rb) revealed enhanced ERK1/2 phosphorylation and hypophosphorylation of Rb in the HCC xenografts, thereby confirming the role of OIS in TSPAN5-knockdown-mediated anti-tumor effects (Figure 7F,G). These data suggest TSPAN5 as a promising new drug target for HCC treatment, inducing OIS in DLC1-deficient HCC.

## 4. Discussion

Although HCC is the second leading cause of cancer mortality in the world, and loss of the tumor suppressor deleted in liver cancer 1 (DLC1) occurs in 50% of liver cancers [47], the signaling pathways downstream of DLC1 that mediate liver carcinogenesis are relatively poorly understood. We have recently described that the restoration of DLC1 expression induces cellular senescence [8]; however, the target gene(s) mediating the effects of DLC1 on senescence induction remain to be investigated.

In our study, we identified tetraspanin 5 (TSPAN5), versican (VCAN), meprin 1A (MEP1A), histone cluster 1 H2B family member K (HIST1H2BK) and N-cadherin (CDH2) as novel candidate genes upregulated upon DLC1 depletion. Amongst these, we characterized TSPAN5 as the most interesting candidate, because TSPAN5 was responsible for mediating the effect of DLC1 on HCC cell proliferation. TSPAN5 belongs to the TSPAN superfamily of integral membrane proteins that possess four highly conserved transmembrane proteins and organize laterally to form tetraspanin-enriched microdomains, emerging as key players in the progression of metastasis of several cancers [48]. TSPAN was recently described in hepatic gene networks in obese patients with nonalcoholic fatty liver disease (NAFLD) [49]. Hepatic accumulation of free fatty acids and oxysterols facilitates the development and progression of NAFLD into steatohepatitis (NASH) [3]. NASH has been shown to increase the risk of developing HCC due to the growing epidemic of obesity and diabetes [4]. We demonstrated the pivotal contribution of TSPAN5 to HCC expansion, as TSPAN5 expression in HCC tissue was markedly increased compared to non-tumorous tissues. Consistent with our results, recent analyses of the Oncomine public database have shown that TSPAN5 is highly expressed in liver and colon cancer [50]. Indeed, we found that HCC cell proliferation, migration, invasion and spheroid formation was almost completely abolished upon TSPAN5 depletion in HCC cells. Most importantly, TSPAN5 knockdown was also able to suppress HCC tumor growth in a CAM and in a xenograft model in vivo. These results indicate TSPAN5 as a novel molecular target for therapeutic intervention of DLC1-deficient HCC. The delivery of intact siRNA is a major bottleneck of siRNA therapies. For providing the first evidence that knockdown of TSPAN5 can be used as a therapeutic approach in HCC tumor xenografts, we used a PEI-based delivery platform for TSPAN5 siRNA, thus allowing a therapeutic intervention after the establishment of the tumors. Ten micrograms of siRNA per injection was selected, since this amount has been found previously to be efficient for inducing profound gene knockdown in tumor xenografts without adverse effects due to the polymer (non-specific PEI effects) or the siRNA; see, e.g., [51]. It should be noted that this dosage (~0.4 µg/kg body weight) is rather low, especially when considering that the complexes were applied systemically, thus also demonstrating the efficacy of siRNA delivery and of siTSPAN5 effects on HCC tumor growth. Previous biodistribution studies had demonstrated intraperitoneal (i.p.) injections to be superior over intravenous (i.v.) injections for delivering siRNA into s.c. tumor xenografts [51]. Upon i.p. injection, PEI/siRNA complexes become systemically available, thus reaching distant organs and tissues, such as the tumors. While i.v.-injected complexes are mostly delivered into the liver, lung and spleen, i.p. injections are superior in regard to reaching the tumor, and were thus preferred here. Efficient TSPAN5 knockdown in vivo achieved by this PEI-based delivery platform for TSPAN5 siRNA reduced tumor growth by driving HCC cells into senescence. Thus, blocking TSPAN5 may be a promising targeted strategy for personalized HCC therapy of patients lacking DLC1. The collated data are consistent with a model in which the tumorigenic effects of DLC1-induced TSPAN5 overexpression are therapeutically explored by inducing anti-tumorigenic and therapeutic effects upon TSPAN5 depletion (Figure 8).

Besides siRNA therapies, monoclonal antibody therapy is an effective regimen for cancer. For other tetraspanins, such as TSPAN8 and CDC151, antibody-based targeting has been proven to be an effective strategy with which to suppress tumor progression and metastasis [52]. Further studies are needed to evaluate the therapeutic efficacy of newly generated specific and high-affinity antibodies against TSPAN5 for HCC therapy [53]. 

On the molecular level, we observed the recruitment of the transcriptional coactivator MRTF-A and the actin-binding protein FLNA to the TSPAN5 promoter. These data further substantiate our previous studies on MRTF-A emerging as key regulator of hepatocarcinogenesis and the senescence response, and on FLNA being required for MRTF-A transcriptional activity [8,14,23,24]. Since the promoters of the other gene targets induced by DLC1 loss, VCAN and CDH2, show occupancy of MRTF and FLNA as well, we cannot exclude their possible contribution to the MRTF-A-associated senescence response. However, we found that TSPAN5 was responsible for mediating the effects of DLC1 loss on RhoA activation and MRTF-dependent target gene expression. TSPAN5 expression itself emerges as the key regulatory step in the control of HCC cell proliferation, migration and invasion. TSPAN5 being both downstream and upstream of MRTF-A (Figure 8) implies a positive feedback loop, in which TSPAN5 feeds back to the production of TSPAN5 mRNA to meet needs in the course of cell proliferation, migration and invasion. Consistent with this, TSPAN5 depletion was necessary and sufficient to break up the positive feedback loop and induce OIS via activation of the MAPK cascade and the p16^INK4a^/Rb pathway as well as the manifestation of the marker of constitutive heterochromatin H3K9me3, in vitro and in vivo. Rb has been shown to cooperate with the promyelocytic leukemia protein (PML), the main constituent of spherical nuclear structures known as PML bodies, during the establishment of senescence [54]. Upon TSPAN5 depletion, we observed an accumulation of PML nuclear bodies, in which Rb/E2F complexes can localize together with proteins involved in heterochromatinization during senescence. In recent years, emerging evidence suggests that senescent cells, beyond undergoing cell cycle arrest, also secrete extracellular vesicles such as exosomes, containing secreted proteins referred to as senescence-associated secretomes (SASs), in order to communicate with their microenvironment. SASP factors include cytokines, chemokines and growth as well as adhesion factors with the ability to turn senescent and adjacent cells into pro-inflammatory cells [55]. Exosomes have been shown to be important mediators of this pro-tumorigenic function of senescent cells [56]. Notably, TSPANs represent important factors for exosome biosynthesis [57]. It is therefore conceivable that a defect in exosome biosynthesis, nuclear content loading and SASP factor secretion contributes to the anti-tumor effects of TSPAN5 silencing observed in CAM tumors and HCC xenografts. Recent studies have also demonstrated the presence of genomic DNA shuttled into nuclear exosomes by tetraspanins and revealed that 10% of cancer-cell-derived exosomes carry nuclear contents [58]. 

Taken together, our results may have implications for the diagnosis and therapy of HCC, since the enrichment of tetraspanins in exosomes provides a reliable diagnostic tool for DLC1-deficient HCC. Moreover, the inhibition of TSPAN5 may offer novel therapeutic approaches for the targeted treatment of HCC by exploring senescence-inducing strategies.

## 5. Conclusions

In this report, we have demonstrated that therapeutic knockdown inhibits HCC tumor growth by inducing oncogene-induced senescence via the actin–MRTF axis. This is the first piece of evidence that TSPAN5 is a promising novel therapeutic target for the treatment of HCC characterized by DLC1 loss.

## Figures and Tables

**Figure 1 cancers-13-05373-f001:**
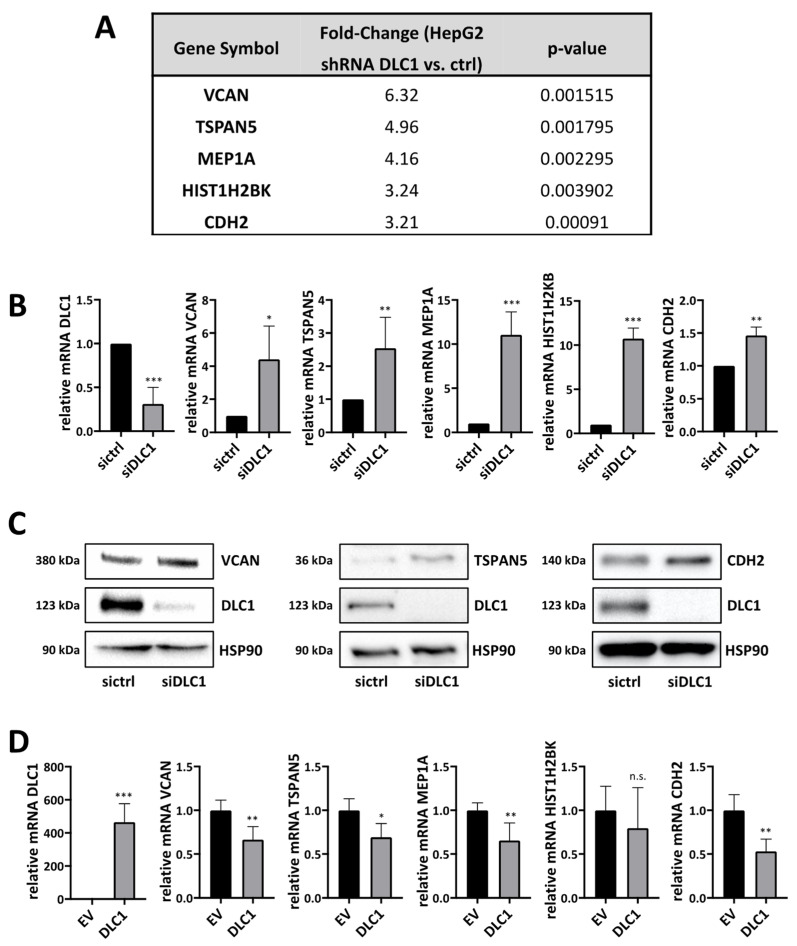
Gene expression profiling to identify novel target genes induced by DLC1 loss. (**A**) DNA microarray analysis was performed to analyze the transcriptome of HepG2 cells either expressing control shRNA (ctrl) or DLC1 shRNA. Genes are ordered to the average fold of upregulation upon DLC1 depletion and the corresponding p-values are given. The threshold for DLC1 dependence was set at 3.2-fold upregulation. (**B**) Expression levels of the indicated target genes in HepG2 cells expressing ctrl siRNA (sictrl) or DLC1 siRNA (siDLC1) and the knockdown efficiency of DLC1 (after 48 h), measured by qRT-PCR using gene-specific primers and normalized to the endogenous housekeeping gene 18S rRNA. Values are means ± SD (*n* = 3); *, *p* < 0.05, **, *p* < 0.01 and ***, *p* < 0.001. (**C**) Lysates of HepG2 cells expressing ctrl siRNA (sictrl) or DLC1 siRNA (siDLC1) were immunoblotted with anti-versican, anti-N-cadherin, anti-tetraspanin 5, anti-DLC1 and anti-HSP90 antibodies. Representative blots of three independent experiments are shown. (**D**) Expression levels of DLC1 and DLC1-dependent genes in HuH7 cells transiently transfected with FLAG-tagged DLC1 (DLC1) or a FLAG-tagged empty vector (EV) were assessed via qRT-PCR as described above. Data are means ± SD (*n* = 3); *, *p* < 0.05, **, *p* < 0.01 and ***, *p* < 0.001. The whole Wester Blots are available at Appendix A.

**Figure 2 cancers-13-05373-f002:**
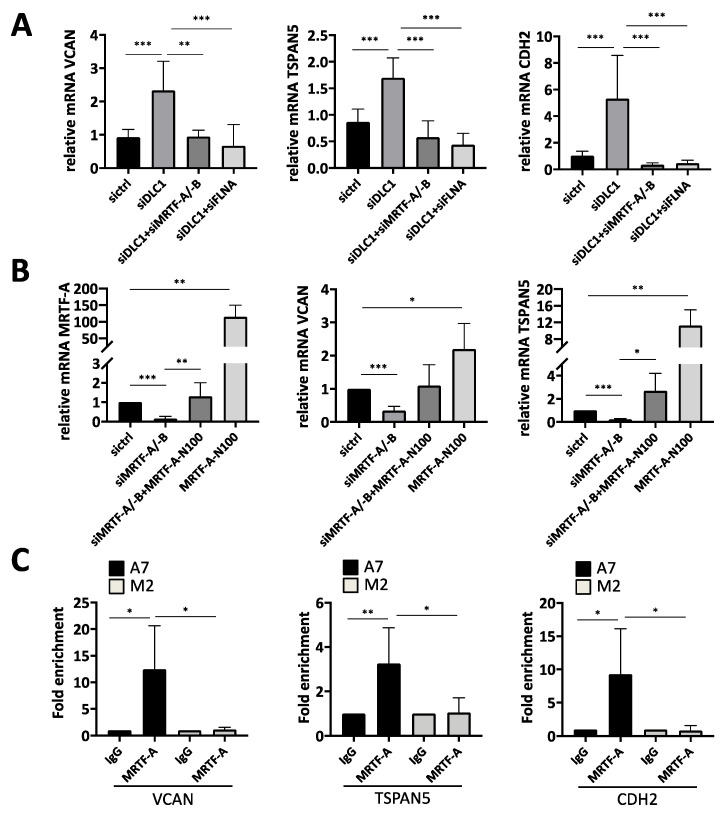
Transcriptional regulation of the newly identified DLC1-dependent genes. (**A**) mRNA expression of TSPAN5, VCAN and CDH2 in HepG2 cells transiently transfected with ctrl siRNA (sictrl), DLC1 siRNA (siDLC1) and a combination of DLC1 siRNA (siDLC1) and MRTF-A/-B siRNA (siMRTF-A/-B) or a combination of DLC1 siRNA (siDLC1) and FLNA siRNA (siFLNA) was assessed by qRT-PCR. The respective gene-specific primers were used and normalization to the 18 S rRNA was carried out. Values are means ± SD (*n* = 3); **, *p* < 0.01 and ***, *p* < 0.001. (**B**) Expression of MRTF-A, VCAN and TSPAN5 mRNA in HuH7 cells after RNAi-mediated MRTF-A/-B knockdown (siMRTF-A/-B), followed by ectopic expression of MRTF-A-N100 for 3 days, determined by qRT-PCR as described above. Values are means ± SD (*n* = 3); *, *p* < 0.05, **, *p* < 0.01 and ***, *p* < 0.001. (**C**) ChIP was performed in three independent chromatin preparations from FLNA-expressing A7 cells compared to FLNA-deficient M2 cells with MRTF-A and IgG antibodies for pulldown. The specific primers for *VCAN*, *TSPAN5* and *CDH2* promoters listed in Appendix A were used. Values are means ± SD (*n* = 3); *, *p* < 0.05 and **, *p* < 0.01.

**Figure 3 cancers-13-05373-f003:**
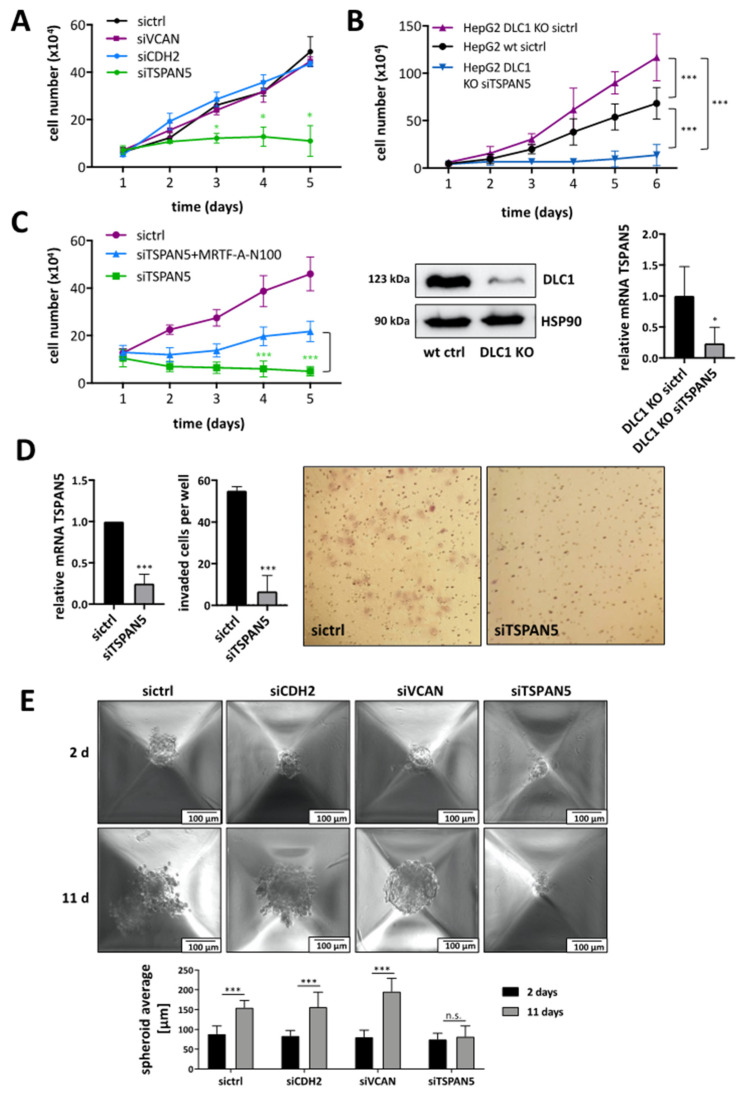
Requirement of TSPAN5 for HCC proliferation, migration, invasion and spheroid formation. (**A**) HuH7 cells transiently transfected with negative control (sictrl) and TSPAN5, CDH2 or VCAN siRNA (siTSPAN5, siCDH2 and siVCAN) were counted daily for 5 days. Values are means ± SD (*n* = 3); *, *p* < 0.05. (**B**) HepG2 DLC1 wt (HepG2 wt) cells transiently transfected with negative control (sictrl) as well as HepG2 DLC1 KO cells transiently transfected with negative control (sictrl) and TSPAN5 siRNA (siTSPAN5) were counted daily for 5 days. Values are means ± SD (*n* = 3); *** *p* < 0.001. Lysates of HepG2 DLC1 wt (wt ctrl) and HepG2 DLC1 KO (DLC1 KO) cells were immunoblotted with anti-DLC1 and anti-HSP90 antibodies to demonstrate the DLC1 KO. Representative blots of three independent experiments are shown. TSPAN5 knockdown efficiency was determined by qRT-PCR with gene-specific TSPAN5 and 18S rRNA primers. Values are means ± SD (*n* = 3); *, *p* < 0.05. (**C**) HuH7 cells expressing scrambled (sictr) and TSPAN5 siRNA (siTSPAN5) were transiently transfected with MRTF-A-N100 and counted daily for 5 days. Values are means ± SD (*n* = 3). (**D**) HuH7 cells were transiently transfected with scrambled siRNA (sictrl) and TSPAN5 siRNA (siTSPAN5). TSPAN5 knockdown efficiency was determined by qRT-PCR with gene-specific TSPAN5 and 18S rRNA primers. Transfected HuH7 cells were subjected to Matrigel^®^ invasion assay chambers and after 48 h invaded cells were stained by crystal violet and counted. Statistical analysis was performed using the unpaired *t*-test. Values are means ± SD (*n* = 3); *** *p* < 0.001. (**E**) HuH7 transfected with negative control siRNA (sictrl), CDH2 siRNA (siCDH2), VCAN siRNA (siVCAN) and TSPAN5 siRNA (siTSPAN5) were transduced to a special spheroid plate and spheroid formation was monitored for 11 days. The spheroid average of day 2 and day 11 were compared (bottom panel). Values are means ± SD (*n* = 3); *** *p* < 0.001. The whole Wester Blots are available at Appendix A.

**Figure 4 cancers-13-05373-f004:**
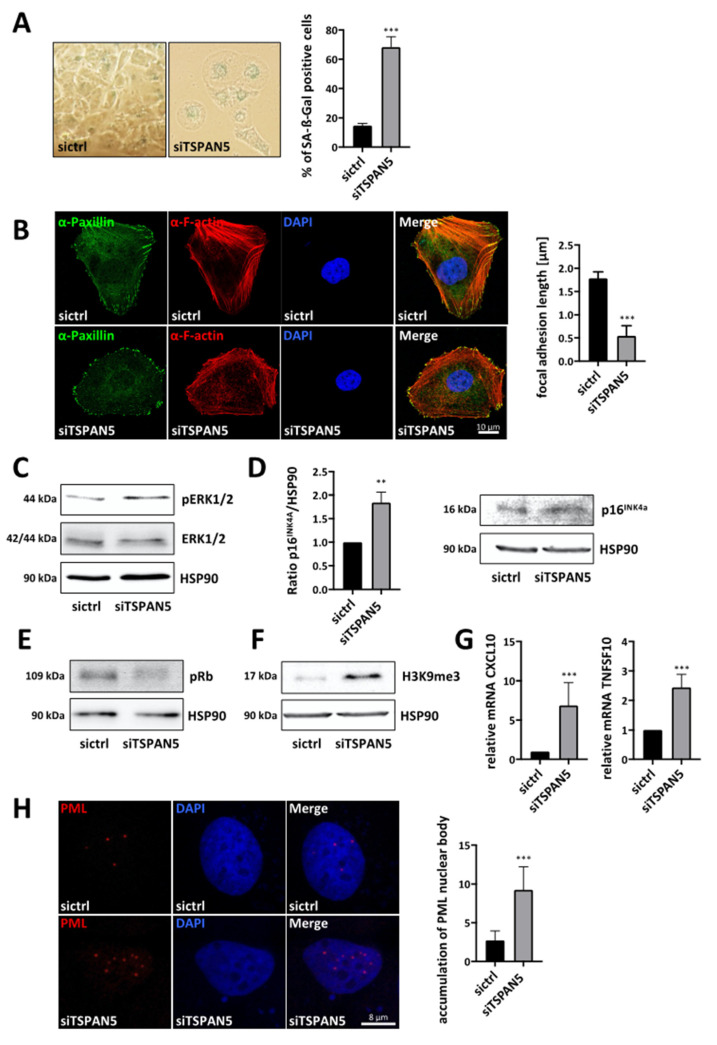
Requirement of TSPAN5 for oncogene-induced senescence. (**A**) Quantification of senescence-associated ß-galactosidase-positive HuH7 cells transiently transfected with control siRNA (sictrl) and TSPAN5 siRNA (siTSPAN5). ß-galactosidase staining was performed 5 days after transfection and ß-gal-positive cells were counted in 100 cells per condition. Data are means ± SD (*n* = 3); ***, *p* < 0.001. (**B**) Immunofluorescence analysis of focal adhesions with α-paxillin staining and F-actin staining with phalloidin in HuH7 cells transfected with TSPAN5 siRNA (siTSPAN5) or negative control siRNA (sictrl). DAPI was used for nuclei staining. The focal adhesion length was measured manually in 50 cells in each of the 5 randomly selected fields by using the ImageJ software in three independent experiments. Scale bar, 10 µm. Data are means ± SD (*n* = 3); ***, *p* < 0.001. (**C**–**F**) Lysates of HuH7 cells transiently transfected with negative control siRNA (sictrl) or TSPAN5 siRNA (siTSPAN5) for 6 days were immunoblotted with anti-pERK1/2 and anti-ERK (**C**), anti-p16^INK4a^ (quantitation on the left) (**D**), anti-pRb (**E**) and anti-H3K9me3 antibodies (**F**). HSP90 was used as a loading control and immunoblotted with anti-HSP90 antibodies. (**G**) HuH7 cells transfected with negative control (sictrl) or TSPAN5 siRNA (siTSPAN5) were subjected to qRT-PCR with the respective gene-specific primers and 18S rRNA primers for normalization in order to analyze *CXCL10* and *TNFSF10* mRNA expression. TSPAN5 knockdown efficiency is shown in Appendix A. Values are means ± SD (*n* = 3); ***, *p* < 0.001. (**H**) Immunofluorescence staining with anti-PML antibodies and DAPI for nuclear counterstaining in HuH7 cells expressing control (sictr) or TSPAN5 siRNA (siTSPAN5) (left). Three independent experiments were performed. Scale bar, 5 µm. Quantification of PML nuclear body accumulation by counting the red spots in 50 cells per condition. DAPI was used for nuclei staining. Data are means ± SD (*n* = 3); ***, *p* < 0.001, **, *p* < 0.01. The whole Wester Blots are available at Appendix A.

**Figure 5 cancers-13-05373-f005:**
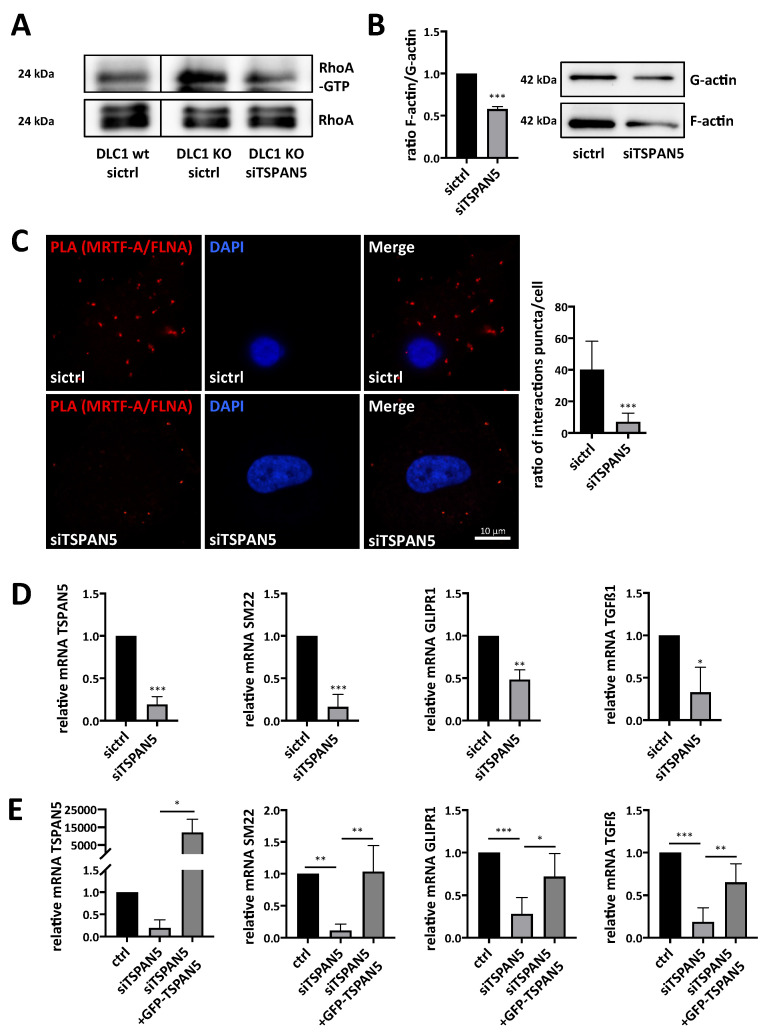
Depletion of TSPAN5 affects the actin/MRTF signaling axis. (**A**) Lysates from HepG2 DLC1 wt (DLC1 wt) and HepG2 DLC1 KO (DLC1 KO) cells transiently transfected with TSPAN5 siRNA (siTSPAN5) or control siRNA (sictrl) were immunoprecipitated with anti-active RhoA antibodies and immunoblotted with anti-RhoA antibodies. Lysates with an equal amount of total protein were directly immunoblotted with anti-RhoA antibodies. The knockdown efficiencies of DLC1 and TSPAN5 are shown in Figure 3B. (**B**) Immunoblotting with anti-actin antibodies and quantification of the ratio of F/G-actin in HuH7 cells transiently transfected with TSPAN5 siRNA (siTSPAN5) as compared to negative control siRNA (sictrl). Data are means ± SD (*n* = 3); ***, *p* < 0.001. (**C**) Immunofluorescence analysis and quantification of proximity ligation assay (PLA) for endogenous MRTF-A and FLNA in HuH7 cells transiently transfected with TSPAN5 siRNA (siTSPAN5) compared to negative control siRNA (sictrl). Scale bar, 10 μm. PLA signals were counted in 15 cells per condition. All data are means ± SD (*n* = 3); ***, *p* < 0.001. (**D**) MRTF target gene expression in HuH7 cells transfected with TSPAN5 siRNA (siTSPAN5) and scrambled siRNA (sictrl), determined by qRT-PCR using SM22-, GLIPR1- and TGFß1-specific primers and 18S rRNA primers for normalization. Values are means ± SD (*n* = 3); *, *p* < 0.05, **, *p* < 0.01 and ***, *p* < 0.001. (**E**) MRTF target gene expression in HuH7 cells treated as above and transfected with a GFP-empty vector (ctrl) or GFP-TSPAN5. Values are means ± SD (*n* = 3); *, *p* < 0.05, **, *p* < 0.01 and ***, *p* < 0.001. The whole Wester Blots are available at Appendix A.

**Figure 6 cancers-13-05373-f006:**
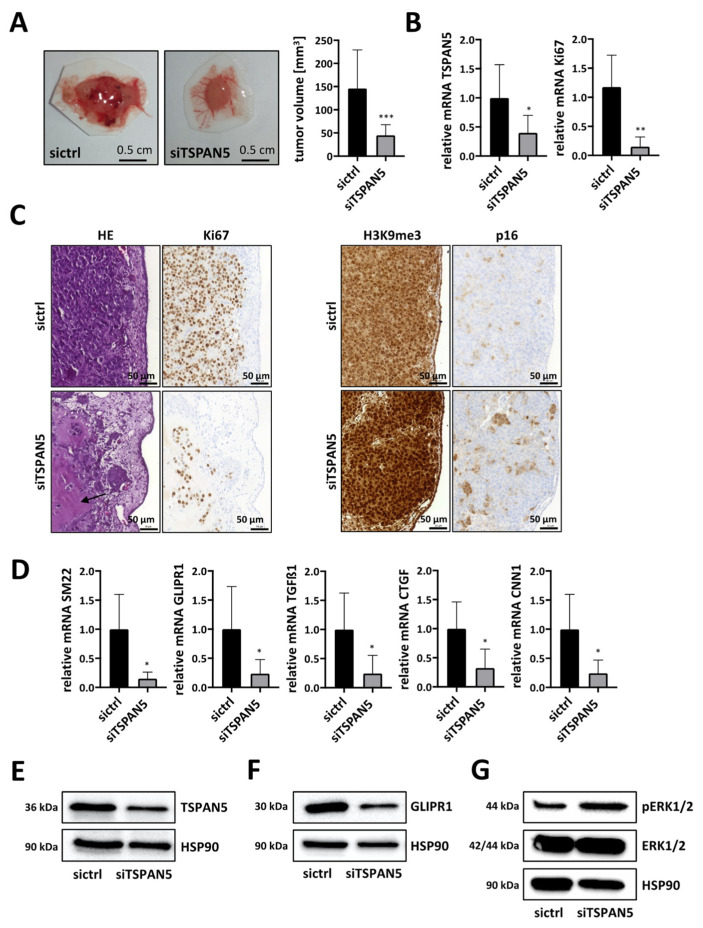
Anti-tumor effects of TSPAN5 depletion in the CAM assay. (**A**) HuH7 cells transfected with negative control siRNA (sictrl) and TSPAN5 siRNA (siTSPAN5) in Matrigel^®^ pellets were transferred to the CAM of fertilized chicken eggs. Five days later tumor pellets of HuH7 cells were extracted, ex ovo images of micro-tumors were taken and tumor volume was calculated. Data are presented as means ± SD (*n* = 17); ***, *p* < 0.001. (**B**) Knockdown efficiency of TSPAN5 and Ki67 expression in micro-tumors generated in (**A**), determined by qRT-PCR using TSPAN5, Ki67 and 18S primers as endogenous housekeeping genes for normalization. Values are means ± SD (*n* = 6); *, *p* < 0.05 and **, *p* < 0.01. (**C**) Left panel: representative photomicrographs of hematoxylin–eosin (HE)- and Ki67-stained paraffin sections of HuH7 sictrl and siTSPAN5 CAM tumors generated as in (**A**). Arrowhead: Undigested Matrigel. Scale bar = 50 µm. Right panel: Representative images of H3K9me3- and p16-stained paraffin sections of HepG2 sictrl and siTSPAN5 CAM tumors prepared as in (**A**). Scale bar = 50 µm. (**D**) RNA was purified from extracted micro-tumors developed from Matrigel and HuH7 cells treated with scrambled RNA (sictrl) or TSPAN5 siRNA (siTSPAN5) in ovo for 5 days and MRTF target gene expression analyzed by qRT-PCR as described in (**B**). Values are means ± SD (*n* = 6); *, *p* < 0.05. (**E**–**G**) Lysates from extracted micro-tumors grown from HuH7 sictrl and HuH7 siTSPAN5 cells in ovo for 5 days as in (**A**) were immunoblotted with anti-TSPAN5 (**E**), anti-GLIPR1 (**F**), anti-ERK^pT202/pY204^, anti-ERK (**G**) or anti-HSP90 antibodies as a loading control. The whole Wester Blots are available at Appendix A.

**Figure 7 cancers-13-05373-f007:**
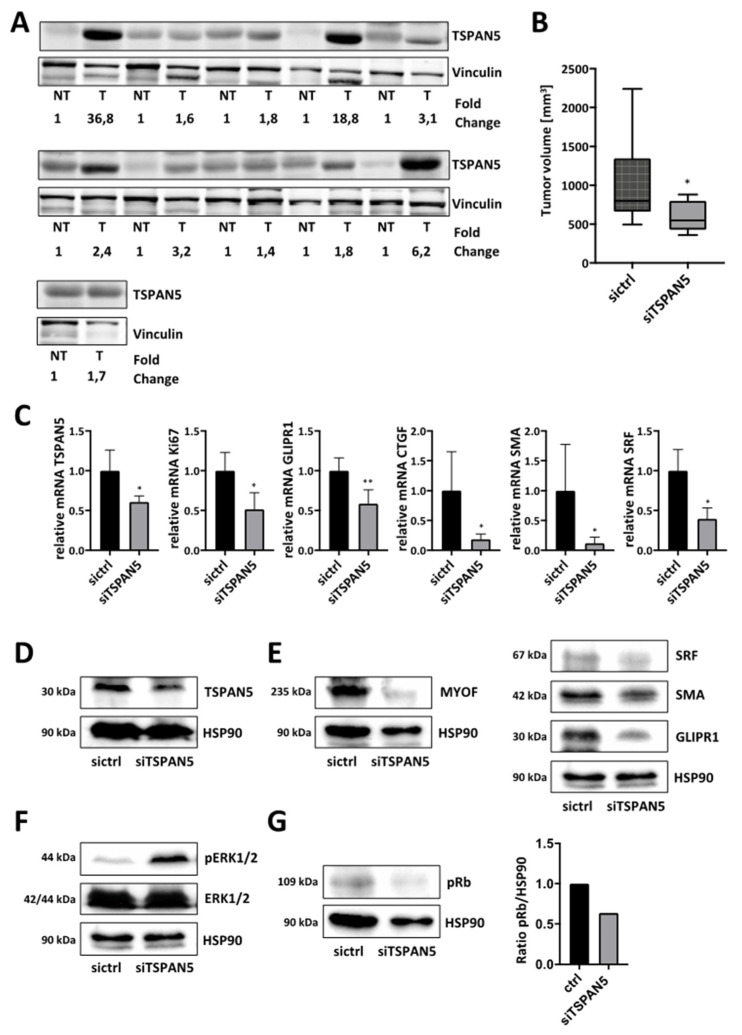
Anti-tumor effects of TSPAN5 depletion in xenografts. (**A**) Tissue samples of HCC patients were immunoblotted with anti-TSPAN5 antibodies and anti-vinculin antibody as a loading control. Relative fold changes of TSPAN5 protein expression (normalized to vinculin) between each pair of corresponding non-tumorous (set to 1) and tumorous tissue are indicated. (**B**) HuH7 HCC-xenograft-bearing mice were randomized and treated on day two, four, six and eight by systemic injection with polymeric nanoscale complexes containing TSPAN5 siRNA (siTSPAN5) or ctrl siRNA (sictrl). Tumor growth was measured manually. Means of tumor sizes are shown for the control and treated groups after day nine of treatment start. Data are presented as means ± SD (*n* = 6 mice per group); *, *p* < 0.05. (**C**) *TSPAN5* knockdown efficiency, *Ki67* mRNA expression and MRTF target gene expression of *GLIPR1*, *CTGF*, *SMA* and *SRF* in HCC xenografts after treatment with PEI/TSPAN5 siRNA (siTSPAN5) vs. PEI/control siRNA (sictrl), determined by qRT-PCR using the respective primers and 18S rRNA as an endogenous housekeeping gene for normalization. All data are means ± SD (*n* = 5); *, *p* < 0.05 and **, *p* < 0.01. (**D**–**G**) Lysates of HCC xenografts treated as above were immunoblotted with anti-TSPAN5 (**D**), anti-MYOF, anti-SRF, anti-GLIPR1 (**E**), anti-ERK^pT202/pY204^ and anti-ERK (**F**), anti-pRb (**G**) and anti-HSP90 antibodies. The whole Wester Blots are available at Appendix A.

**Figure 8 cancers-13-05373-f008:**
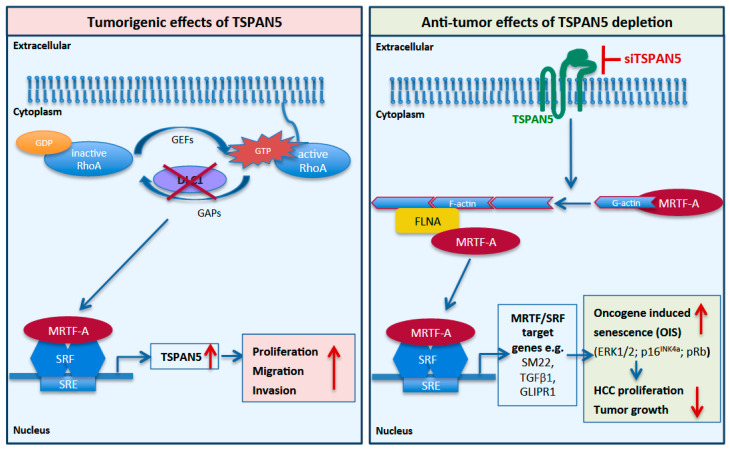
Model for the tumorigenic effects of TSPAN5 and the anti-tumorigenic effects of TSPAN5 depletion. DLC1 loss leads to increased TSPAN5 expression via MRTF-A transcriptional activation (left panel). Therapeutic knockdown of TSPAN5 results in OIS-induced inhibition of HCC growth via decreased actin polymerization and reduced MRTF/SRF target gene expression (right panel).

## Data Availability

The data presented in this study are available in this article (and Appendix A).

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
