# Peer review of "Tetraspanin 5 (TSPAN5), a Novel Gatekeeper of the Tumor Suppressor DLC1 and Myocardin-Related Transcription Factors (MRTFs), Controls HCC Growth and Senescence"

_cancers, 2021, doi:10.3390/cancers13215373_

Round 1

Reviewer 1 Report

In this manuscript, Schreyer et al. used gene expression profiling to investigate changes in gene expression upon loss of the tumor suppressor DLC1 in human hepatoma cells and identified a set of upregulated target genes upon DLC1 depletion. The authors show that this regulation is dependent on FLNA and the transcriptional co-activator MRTF-A, which they identified previously as regulators of hepatocarcinogenesis after DLC1 loss. Out of this set of genes, the authors focus on the transmembrane protein Tetraspanin 5 (TSPAN5) and convincingly show that depletion of TSPAN5 in DLC1-negative cells leads to defects in cell growth, invasion and migration, and the induction of senescence in vitro as well as in CAM and mouse HCC xenografts models in vivo. Thus, this study shines further light on molecular consequences downstream of DLC1 loss and identifies TSPAN5 as a potential therapeutic target in HCC.  However, some issues need to be addressed before the study is ready for publication:

Major

General: While the authors nicely demonstrate that TSPAN5 mediates proliferation, migration and invasion of DLC1-negative HCC cells in various models, there is no experiment showing that TSPAN5 is responsible for mediating the effects of DLC1 loss. This would require knocking down DLC1 and TSPAN5 simultaneously in HepG2 cells followed by the analysis of cellular consequences, instead the authors only knock down MRFT-A to establish TSPAN5 as a transcriptional target.

Figure 5 – Mechanistically it is not clear how TSPAN5 mediates the effects on actin/MRFT‑A.  Moreover, with TSPAN5 being both downstream and upstream of actin/MRFT-A, does this imply a positive feedback?

Figure 7 – Does the TSPAN5 status inversely correlate with the DLC1 status in HCC tissues? The membrane should be reprobed for DLC1 expression levels. Further analysis of TCGA expression data should be performed to validate the in vitro results in larger patient cohorts and establish TSPAN5 as a biomarker for DLC1-negative HCC.

Minor

Figure 4D: Due to high background the increase in p16INK4A is not immediately obvious. Could the authors provide a better blot or quantification?

Figure 7C-G: Can the authors clarify why one of the six xenografts was excluded from the mRNA analysis? Did they use this single tumor for immunoblotting analysis or did they confirm the effect on the protein level in multiple xenografts?

Can the authors comment on targeting TSPAN5 via antibodies as a therapeutic approach?

Page 5, line 252: cells were transduced with DLC1 shRNA (to) for RNAi induction

Figure 4: the staining is α-F-actin (using Phalloidin), not α-Phalloidin.

Reviewer 2 Report

HCC is one of the most devastating malignancies and is the second leading cause of cancer-related death worldwide. Therefore, urgent need to find an underlying mechanism involved in HCC growth and proliferation. Tetraspanins belong to a transmembrane domains protein family, widely reported in many cancers' biological functions, including HCC. In this manuscript, the authors demonstrate Tetraspanins 5 (TSPAN5), a novel gatekeeper of the tumor suppressor DLC1 and Myocardin-related transcription factor 4 (MRTF), controls HCC growth and senescence. They provide several positive outcomes from this study. However, I have few concerns.

  1. The name of cell lines should be listed in material and methods.
  2. Most methods such as cell proliferation assay, immunoblotting, source of primer synthesis, number of mice per group, etc., are not clear that need to be re-write.
  3. What is the rationale for selecting Huh7 cells for the xenograft model?
  4. The basis for deciding the dose of PEI F25-LMW/siRNA complexes containing 10 μg siRNA; and why intraperitoneal (i.p.) was used while i.v. injection is the best route for drug administration?
  5. What methods were used to sacrifice the mice?
  6. The authors demonstrated HCC proliferation, migration, invasion, spheroid formation, oncogene-induced senescence, MTRF signaling, and CAM assay in Huh7 cells only. They did not show these studies in another cell line, HepG2. The authors should be transparent with those studies in both cell lines to understand the difference in the mechanism.

Reviewer 3 Report

Very interesting study. I just recommend to expand in the discussion the comments on the general concepts of hepatocarcinogenesis. In this regard, cite two studies (PMID: 29220698; PMID: 23845075)

Author Response

We thank the reviewer and agree that it is very important to note that 50% of patients with HCC suffer from metabolic syndrome such as obesity or diabetes, and that recent studies demonstrated that type II diabetes confers a three fold risk of HCC. We included the references and the link between NAFLD, NASH and HCC in the discussion on page 13.

TSPAN was recently decribed in hepatic gene networks in obese patients with Nonalcoholic Fatty Liver Disease (NAFLD) [49]. Hepatic accumulation of free fatty acids and oxysterols facilitates the development and progression of NAFLD into steatohepatitis (NASH) [3]. NASH has been shown to increase the risk of developing HCC due to the growing epidemic of obesity and diabetes [4].

Reviewer 4 Report

The manuscript "Tetraspanin 5 (TSPAN5), a novel gatekeeper of the tumor suppressor DLC1 and Myocardin related transcription factor (MRTF), controls HCC growth and senescence" by Schreyer et al. provides evidence on the role of tetraspanin 5 on hepatocellular carcinoma. The study is well-designed and the manuscript is well-writen. The provided data are convincing and support the conceptual framework of the proposed mechanism.

Therefore, I suggest the acceptance of the manuscript in its current version. 

Author Response

Thank you!

Reviewer 5 Report

The authors conducted a study to explore the downstream target of the tumor suppressor gene DLC1 in HCC. The study has novelty and provides meaningful data.

My comments:

1. The authors constantly shifted between various HCC cell lines in different experiments. Please at least show complete data in at least one cell line. For example, if most phenotype and mechanistic studies were done in HuH7 cells, then please provide Figures 1 and 2 in HuH7 cells as well.

2. Fig. 2B. Only results with VCAN and TSPAN5 were shown. How about CDH2? 

3. Many of the key experiments were only performed with one siRNA for one gene. Because siRNA is common for off-target effects, the authors should test with another set of siRNA for important data. For example, TSAPN5 is the key target in this study. Throughout the entire study, only one siRNA of TSPAN5 was used in experiments.

4. For the xenograft study, the authors should provide tumor growth curves by day, not just the end tumor volume.

Round 2

Reviewer 2 Report

The authors have addressed all the concerns raised. The revised version of the manuscript can be considered for publication. 

Author Response

Thank you!

Reviewer 5 Report

The authors adequately replied most of my concerns except one.

I asked for CDH2 data in Figure 2B. The authors refused to do so and said it is out of the scope of the paper. However, both Figures 2A and 2C had CDH2 data. Obviously the authors themselves thought CDH2 data are necessary. Please provide CDH2 data in Figure 2B. Otherwise, it is odd to keep CDH2 in Figures 2A and 2C.

Author Response

We apologize that we didn’t include CDH2 data in the first round of the revision. We inserted now CDH2 data in Supplementary Fig. S2A and added the following sentences on page 8:

Furthermore, endogenous VCAN, TSPAN5 and CDH2 mRNA expression was downregulated upon MRTF-A/-B knockdown, while levels were rescued upon co-transfection of constitutively active MRTF-A (MRTF-A N100) (Fig. 2B, Fig. S2A).

Round 3

Reviewer 5 Report

The authors have replied my comments adequately.